# Inactivation of the Akt/FOXM1 Signaling Pathway by Panobinostat Suppresses the Proliferation and Metastasis of Gastric Cancer Cells

**DOI:** 10.3390/ijms22115955

**Published:** 2021-05-31

**Authors:** Na-Ri Lee, Da-Yeah Kim, Hua Jin, Ruoyu Meng, Ok Hee Chai, Seong-Hun Kim, Byung-Hyun Park, Soo Mi Kim

**Affiliations:** 1Division of Hematology/Oncology, Department of Internal Medicine, Jeonbuk National University Medical School, Jeonju 54907, Korea; nariflower@jbnu.ac.kr; 2Research Institute of Clinical Medicine, Jeonbuk National University, Biomedical Research Institute of Jeonbuk National University Hospital, Jeonju 54907, Korea; 3Department of Physiology, Institute of Medical Science, Jeonbuk National University Medical School, Jeonju 54907, Korea; kdyeah@jbnu.ac.kr (D.-Y.K.); kathymeng1216@gmail.com (R.M.); 4School of Pharmaceutical Sciences, Tsinghua University, Beijing 100084, China; jinhuaxy@126.com; 5Department of Anatomy, Institute of Medical Science, Jeonbuk National University Medical School, Jeonju 54907, Korea; okchai1004@jbnu.ac.kr; 6Department of Internal Medicine, Division of Gastroentrology, Jeonbuk National University Medical School, Jeonbuk National University Hospital, Jeonju 54907, Korea; shkimgi@jbnu.ac.kr; 7Department of Biochemistry, Jeonbuk National University Medical School, Jeonju 54907, Korea; bhpark@jbnu.ac.kr

**Keywords:** panobinostat, gastric cancer cells, apoptosis, cell cycle, FOXM1

## Abstract

Gastric cancer is the fifth most common cancer and the third leading cause of cancer-related deaths worldwide. Histone deacetylase (HDAC) inhibitors are a new class of cytostatic agents available for the treatment of various cancers and diseases. Although numerous clinical and pre-clinical trials on the anticancer effects of panobinostat have been conducted, only a few reports have investigated its efficacy in gastric cancer. The present study aimed to investigate the effects of panobinostat in gastric cancer cells. Panobinostat significantly inhibited the cell viability and proliferation of the gastric cancer cell lines SNU484 and SNU638 in a dose-dependent manner; it reduced the colony-forming ability of these cells. Moreover, it induced apoptosis as indicated by increased protein levels of cleaved poly ADP-ribose polymerase and cleaved caspase-3. Panobinostat induced the G2/M cell cycle arrest in SNU484 and SNU638 cells and subsequently decreased the G2/M phase regulatory-associated protein expression of p-Wee1, Myt1, and Cdc2. Furthermore, panobinostat significantly inhibited the metastasis of SNU484 and SNU638 cells by regulating the expression of MMP-9 and E-cadherin. Further, it decreased the protein levels of p-Akt and forkhead box protein M1 (FOXM1). These effects were reversed by the Akt agonist SC79 and were accelerated by the Akt inhibitor LY2940002. Moreover, tumor growth in xenograft animal experiments was suppressed by panobinostat. These results indicated that panobinostat inhibits the proliferation, metastasis, and cell cycle progression of gastric cancer cells by promoting apoptosis and inactivating Akt/FOXM1 signaling. Cumulatively, our present study suggests that panobinostat is a potential drug for the treatment of gastric cancer.

## 1. Introduction

Gastric cancer (GC) is the fifth most common cancer and the third leading cause of cancer-related mortality worldwide [1]. GC is a major health concern, with over 1 million new cases globally each year [2]. Moreover, owing to metastatic recurrence, the prognosis for patients with advanced GC remains poor [3,4]. Therefore, the development of novel chemotherapies is urgently required to control GC progression and recurrence.

Histone deacetylase (HDAC) inhibitors exhibit their tumor-suppressive effects by increasing the acetylation of histones and non-histone proteins, thereby regulating gene transcription and leading to the induction of apoptosis, differentiation, and degradation of misfolded proteins [5]. Epigenetic modulations frequently occur in cancer and play an important role in GC pathogenesis. Therefore, histone modifications may influence aberrant gene expression profiles and affect tumor suppressor gene silencing [6]. In recent years, various classes of HDAC inhibitors have been investigated for their potential antitumor effects. Vorinostat is approved for the treatment of advanced primary cutaneous T-cell lymphoma [7], and belinostat is approved for the treatment of relapsed or refractory peripheral T-cell lymphoma [8]; other HDAC inhibitors are currently being investigated in clinical and pre-clinical studies. The hydroxamic acid-based HDAC inhibitor panobinostat is produced and marketed by Novartis for the treatment of various cancers [9]. In February 2015, panobinostat was approved by the US Food and Drug Administration as treatment for patients with multiple myeloma who have received at least two previous regimens, including bortezomib and an immunomodulatory agent [10,11], and by the European Medicines Agency for the same use [12]. To date, the anticancer effects of panobinostat have been examined in breast cancer [13], lung cancer [14], prostate cancer [15], and other cancer types. Although one study reported that the HDAC inhibitor LBH589 (panobinostat) sensitizes GC cells via the induction of CITED2 (Regel, 2012 #862), the detailed mechanism of panobinostat on GC remains unelucidated.

Forkhead box protein M1 (FOXM1) is a pivotal proliferation-associated transcription factor [16]. FOXM1 is a crucial regulator of the G2/M phase cell cycle transition and mitotic spindle integration [17,18] and suggested as a key player in tumorigenesis [16,19]. The elevated expression of FOXM1 was observed in several aggressive human cancers, including breast cancer [20], hepatocellular carcinoma [21], colorectal cancer [22], and GC [23]. Moreover, the overexpression of FOXM1 enhances chemoresistance in various cancers [24,25,26], whereas the deficiency of FOXM1 suppresses cancer cell proliferation, growth, progression, and oncogenesis [16,19,27,28]. Moreover, FOXM1 is a known downstream factor of the Akt signaling cascade [29] and plays an important role in cancer cell growth and metastasis [30]. However, despite the importance of FOXM1 in GC progression, few studies have examined panobinostat and its modulation of FOXM1 in GC. Furthermore, the precise mechanism via which panobinostat regulates the Akt/FOXM1 signaling pathway in GC remains unknown. Therefore, the present study aimed to determine whether panobinostat is involved in the metastasis and inhibition of GC cells and to investigate if the effects of panobinostat are mediated by the Akt/FOXM1 signaling pathway. Our results indicated that panobinostat suppresses cell proliferation, cell cycle progression, cell migration, and cell metastasis via inactivating Akt/FOXM1 signaling in GC, suggesting that panobinostat offers a novel treatment strategy for GC.

## 2. Results

### 2.1. Inhibition of Cell Growth and Proliferation by Panobinostat

As an HDAC inhibitor, panobinostat might affect histone acetylation, deacetylation, and/or hyperacetylation. The precise epigenetic modulations by panobinostat treatment were investigated in the present study. Panobinostat induced histone acetylation in a dose-dependent manner, however, no change was observed with histone H3 and H4 in SNU484 and SNU638 GC cells (Figure 1A). SNU484 and SNU638 cells were treated with 0, 10 50, and 100 nM of panobinostat for 48 h. Panobinostat significantly inhibited SNU484 and SNU638 cell viability in a dose-dependent manner at 48 h (Figure 1B). The half maximal inhibitory concentration (IC_50_) of panobinostat was ~50 nM in SNU484 cells and 100 nM in SNU638 cells. Considering the anti-proliferation effect of panobinostat, we performed a colony formation assay. Panobinostat treatment significantly inhibited the colony formation ability of SNU484 and SNU638 cells (Figure 1C). Based on these data, we suggest that panobinostat exhibits potent antitumor effects on GC cells.

### 2.2. Induction of Apoptosis by Panobinostat

To investigate the apoptotic effect of panobinostat, annexin V–FITC/PI staining assay was performed (Figure 2A). The number of annexin V-positive cells was significantly increased following treatment with 100 nM of panobinostat in both cell lines. To further investigate the apoptotic effects of panobinostat on SNU484 and SNU638 cells, the expression levels of apoptosis-associated proteins, including cleaved PARP, PARP, cleaved caspase-3, and caspase-3, were measured using Western blot (Figure 2B). Panobinostat treatment at the indicated doses significantly decreased the expression of PARP and caspase-3 and significantly increased that of cleaved PARP and cleaved caspase-3. Moreover, the proportions of SNU484 and SNU638 cells in the sub-G1 phase were determined using PI staining. Panobinostat at concentrations of 50 and 100 nM significantly increased the proportion of SNU484 and SNU638 cells in the sub-G1 phase (Figure 2C). These results revealed that panobinostat dose-dependently induced apoptotic cell death in SNU484 and SNU638 cells.

### 2.3. Cell Cycle Regulation by Panobinostat

To elucidate whether panobinostat causes growth inhibition in SNU484 and SNU638 cells, cell cycle distribution was determined using fluorescence-activated cell sorting (FACS) analysis. The fraction of cells in each phase of the cell cycle was measured at 48 h after panobinostat treatment (0, 10, 50, and 100 nM). The scatter plots showed that the proportion of cells in the G1 phase was dose-dependently decreased by panobinostat. There was a population increase of 15% of SNU484 cells and 8% proportion of SNU638 cells in the G2/M phase as a result of panobinostat (Figure 3A). Because GC cell accumulation was observed in the G2/M phase, we further investigated whether there were alterations in cell cycle regulatory proteins, such as p-Wee1, Myt1, and Cdc2. Western blotting data suggested that panobinostat arrested SNU484 and SNU638 cells in the G2/M phase by downregulating the expression of p-Wee1, Myt1, and Cdc2 (Figure 3B). The results indicate that panobinostat treatment significantly increases the accumulation of G2/M phase GC cells.

### 2.4. Cell Migration and Metastasis Inhibition by Panobinostat

A wound healing assay was conducted to elucidate the effect of panobinostat on GC cell migration. Panobinostat treatment at the indicated concentrations (0, 10, 50, and 100 nM) significantly decreased cell migration at 24 and 48 h (Figure 4). The migration inhibitory effect was evidently observed at higher concentrations of panobinostat in both cell lines. To further investigate the effect of panobinostat on cell metastasis, a Western blot analysis was performed to elucidate the underlying molecular events. Loss of the protein expression of E-cadherin is a well-established characteristic of invasive epithelial cancers, and MMP-9 plays key roles in regulating local invasion during cancer progression. At 48 h, the protein expression levels of E-cadherin were dose-dependently increased by panobinostat treatment (0, 10, 50, and 100 nM), whereas those of MMP-9 were dose-dependently decreased (Figure 5). Cumulatively, these results suggest that panobinostat suppresses GC cell migration and metastasis.

### 2.5. Downregulation of Akt/FOXM1 Signaling by Panobinostat

To assess the function of panobinostat in the regulation of Akt/FOXM1 signaling, the protein expression levels of Akt, p-Akt, and FOXM1 were measured after panobinostat treatment (0, 10, 50, and 100 nM; Figure 6A). After treatment for 48 h, the protein expression level of p-Akt (activated form of Akt) was effectively decreased, whereas that of total Akt was not significantly changed (Figure 6B). FOXM1 is a critical proliferation-associated transcription factor and a downstream factor of the Akt signaling pathway. Therefore, we measured FOXM1 protein levels after panobinostat treatment. FOXM1 protein expression was significantly decreased by panobinostat treatment in a dose-dependent manner (Figure 6C). To further investigate whether an Akt antagonist can mimic or an Akt agonist can suppress panobinostat functions, we performed the Western blotting using the Akt inhibitor LY2940002 and the Akt agonist SC97 to determine the effect of panobinostat on the protein expression levels of Akt, FOXM1, and MMP-9. LY294002 (10 μM) and SC79 (10 μM) were pre-treated for 2 h, followed by panobinostat treatment (50 nM) for 48 h. LY2940002 combined with panobinostat treatment significantly suppressed the protein expression levels of p-Akt, FOXM1, and MMP-9 in GC cells compared with those observed with panobinostat treatment alone (Figure 6D). Furthermore, SC79 significantly increased the protein expression levels of p-Akt, FOXM1, and MMP-9 by blocking panobinostat in GC cells. These findings indicated that panobinostat significantly downregulates Akt/FOXM1 signaling in a dose-dependent manner in GC cells. Therefore, the results suggest that panobinostat exerts an anticancer effect via the inactivation of Akt/FOXM1 signaling in GC cells.

### 2.6. Inhibitory Effect of Panobinostat on Tumor Growth in an In Vivo Xenograft Animal Model

We further examined whether the anticancer effect of panobinostat is reflected in an in vivo xenograft animal model (Figure 7). SNU484 cells were injected into the right flank of SPF/VAF immunodeficient mice. Xenograft animals were divided into two groups and treated on a daily schedule (5 days on and 2 days off) with either control or panobinostat (10 mg/kg) until the experiment was terminated. The tumor sizes in mice were measured once every 3 days using calipers (Figure 7A). In the panobinostat treatment group, we observed a significant suppression of tumor growth in SNU484 xenograft animals (*p* = 0.007). Moreover, hematoxylin–eosin staining showed notable changes in tumor histology (Figure 7B,C). Aggressively proliferating viable tumor cells with round-to-oval hyperchromatic nuclei were observed in the control group (Figure 7B). However, panobinostat treatment markedly increased the quantity of apoptotic and necrotic cells and the infiltration of lymphocyte around tumor cells (Figure 7C).

## 3. Discussion

The present study was conducted to elucidate the precise mechanism via which panobinostat modulates the Akt/FOXM1 signaling cascade and to determine whether panobinostat is effective as an anticancer agent in GC cells. The effects of panobinostat on Akt/FOXM1 signaling were determined in GC cells. The present results showed that panobinostat suppressed FOXM1 by inactivating Akt. The downregulation of Akt modulated FOXM1 downstream responses, such as cell cycle progression, cell proliferation, cell migration and metastasis, and apoptosis. Therefore, our results suggest that panobinostat reduces GC progression and that it is a potential anticancer agent for the treatment of GC.

Panobinostat, a non-selective HDAC inhibitor, has been examined for its tumor suppressor effect against certain cancers and is currently under investigation in clinical trials for several cancers, such as cutaneous T-cell lymphoma, Hodgkin’s lymphoma, prostate cancer, thyroid cancer, and other solid and hematological malignancies [31,32,33]. Various studies have suggested that panobinostat is closely related to the suppression of cancer progression that is mediated by the regulation of several signaling pathways, including STAT3 [34], EGFR [35], and RAF/ERK [36]. We found that panobinostat effectively inhibited GC cell growth in a dose-dependent manner. Moreover, colony formation ability was significantly suppressed in SNU484 and SNU638 cells. Further, panobinostat dose-dependently promoted the apoptosis of GC cells as indicated by the increased level of c-PARP and cleaved caspase-3, the proportion of sub-G1 cells, and number of annexin V-positive cells. These observations are consistent with those of a previous study, which demonstrated that panobinostat induced apoptosis in acute myeloid leukemia [37]. Therefore, the present study demonstrates that panobinostat induces growth inhibition and apoptosis in GC cells.

Several studies have shown that FOXM1 activation is closely associated with cancer progression [38]. FOXM1—a member of the FOX family of transcription factors—regulates various cell cycle regulatory proteins [21,39,40,41]. Moreover, several studies have demonstrated that high FOXM1 expression is associated with poor prognosis of patients with cancer having various tumors, including colorectal cancer [22], pancreatic ductal adenocarcinoma [42], hepatocellular carcinoma [43], non-small cell lung cancer [43], ovarian cancer [44], and GC [25]. Additionally, FOXM1 is reportedly overexpressed in GC compared with normal gastric epithelium, suggesting that FOXM1 is a prospective diagnostic and prognostic biomarker in GC [45]. Therefore, targeting FOXM1 is a potential therapeutic strategy for the treatment of GC. FOXM1 is a known downstream factor of the Akt signaling pathway that regulates cell survival, proliferation, and metastasis [46]. Therefore, we further investigated the effect of panobinostat on the regulation of the Akt signaling pathway and protein expression of FOXM1. The expression of p-Akt significantly decreased following panobinostat treatment, whereas total Akt was not significantly changed. Moreover, the FOXM1 protein expression level gradually decreased. To determine whether an Akt antagonist and panobinostat have similar functions or whether an Akt agonist can block functions of panobinostat, we examined the protein expression levels of p-Akt, FOXM1, and MMP-9 using the Akt inhibitor LY2940002 and the Akt agonist SC97 in GC cells. LY2940002 with panobinostat treatment significantly reduced the protein expression levels of p-Akt in GC cells compared with those observed with panobinostat treatment alone. Moreover, SC79 treatment significantly increased the protein expression levels of p-Akt, FOXM1, and MMP-9 compared with those observed with panobinostat alone. Based on these results, it can be suggested that panobinostat induces cell death and metastasis via the Akt/FOXM1 signaling pathway. To further investigate the effect of panobinostat on GC metastasis, we performed wound healing assays and measured metastasis-associated protein expression levels. The results indicated an inhibitory effect of panobinostat on cell migration and metastasis. These findings suggest that panobinostat modulates Akt/FOXM1 signaling, which is closely associated with cell growth, proliferation, and metastasis in GC.

Moreover, accumulated studies have shown that that FOXM1 regulates the cell cycle [47,48,49]. FACS analysis was performed to elucidate the alterations in cell cycle distribution, and its results indicated that panobinostat induces the G2/M phase cell cycle arrest in GC cells. Because we observed G2/M phase arrest, we measured the levels of G2/M phase regulatory proteins. These observations were similar to the results of a previous study that demonstrated G2/M cell cycle arrest following corilagin treatment in ovarian cancer [50]. When a previous study investigated the Akt/FOXM1 signaling axis, FOXO3a was shown to be negatively correlated with Akt and FOXM1 [51,52]. Although the expression of FOXO3a was not measured, it is assumed that panobinostat inactivates Akt signaling and increases the accumulation of FOXO3a. This increase in accumulated FOXO3a may inactivate FOXM1 in GC cells. These results suggest that panobinostat may inhibit FOXM1 activation via the suppression of Akt. However, the mechanism in which Akt regulates the distribution of FOXO3a and FOXM1 in the nucleus and cytosol of GC cells by panobinostat, and whether the expression of target genes of FOXM1 can be altered by panobinostat, requires further studies. Therefore, the precise mechanism that panobinostat regulates Akt/FOXO3a/FOXM1 signaling in GC requires further investigation.

## 4. Materials and Methods

### 4.1. Cell Culture and Reagents

The GC cell lines SNU484 and SNU638 were purchased from the Korean Cell Line Bank (Seoul National University, Seoul, South Korea). SNU484 and SNU638 cells were cultured in RPMI 1640 (Gibco, Grand Island, NY, USA) medium supplemented with 10% fetal bovine serum (WELGENE, Gyeongsan-si, South Korea) and 1% penicillin–streptomycin (SIGMA, St. Louis, MO, USA) in a humidified incubator maintained at 37 ℃ with 5% CO_2_. All experiments were performed with cells having ~70–90% confluency. Panobinostat was purchased from ChemCruz from Santa Cruz Biotechnology Inc. (Dallas, TX, USA). LY294002 was procured from Sigma-Aldrich (St. Louis, MO, USA). SC79 was purchased from Tocris Bioscience (Bristol, United Kingdom). Primary antibodies utilized for the cell signaling technology were as follows: GAPDH (#2118s), H3# (9717), H4 (#2592), AcH3 (#8173), AcH4 (#2591), poly ADP-ribose polymerase(PARP) (#9542), cleaved PARP (#9541), caspase-3 (#9662), cleaved caspase-3 (#9664), E-cadherin (#5296), MMP-9 (#13667), phoshpo-Wee1 (#4910), Myt1 (#4282), Akt (9272), phospho-Akt (p-Akt)(#9271), and anti-rabbit IgG (HRP-linked) (#7074) were purchased from Cell Signaling Technology (Denver, MA, USA). FOXM1 (sc-502) and Cdc2 (sc-163) antibodies were purchased from Santa Cruz Biotechnology Inc. (Dallas, CA, USA).

### 4.2. Cell Viability Assay by WST Assay

SNU484 and SNU638 cells were seeded in a 96-well plate (1 × 10^4^ cells/well) containing serum-free medium and incubated overnight. Subsequently, the cells were treated with panobinostat at the indicated concentrations (0, 10, 50, and 100 nM) and dimethyl sulfoxide (DMSO; SIGMA, Saint Louis, MJ, USA) as a negative control. Thereafter, the cells were incubated for 48 h under normal conditions (37 °C, 5% CO_2_, humidified atmosphere). After 48 h, the medium was removed, and the WST assay reagent (EZ-Cytox, DOGEN, Seoul, South Korea) was added according to the manufacturer’s protocol. Absorbance was measured at 450 nm using an Epoch microplate reader (BioTek, Winooski, VT, USA). Over three independent experiments were performed for each cell line.

### 4.3. Colony Formation Assay

SNU484 and SNU683 cells (0.5–2.0 × 10^4^) were seeded in a 6-well plate and incubated overnight. The cells were treated with various concentrations of panobinostat (0, 10, 50, and 100 nM) for 48 h. The cells were incubated for 1–2 weeks. The medium was replaced with fresh medium containing panobinostat twice a week. At the end of the experiment, the medium was discarded, and cells were washed with phosphate-buffered saline (PBS)(Gibco, Grand Island, NY, USA). Colonies were fixed with 100% methanol for 20 min at room temperature. Following fixation, colonies were stained with 0.05% crystal violet (SIGMA, St. Louis, MO, USA) for 5 min. The residual staining solution was removed and washed with 3rd distilled water. The plates were inverted and dried overnight at room temperature. Colonies containing > 50 cells were counted and imaged.

### 4.4. Cell Cycle Analysis

For cell cycle analysis, 1 × 10^6^ SNU484 and SNU638 cells were seeded in 60 mm dishes and incubated overnight under normal conditions (37 °C, 5% CO_2_, humidified atmosphere). The cells were treated at the indicated concentrations (0, 10, 50, and 100 nM) for 48 h. The cells were fixed with 75% ethanol for 2 h at −20℃. The fixed cells were washed with PBS and then incubated with RNase A (10 µg/mL). Following RNase A incubation, the cells were stained with propidium iodide (PI; 10 µg/mL, SIGMA) for 30 min at 37 °C in 5% CO_2_ and humidified conditions. The cellular DNA content was determined using the Accuri C6 flow cytometer (Becton Dickinson, San Jose, CA, USA) and analyzed with BD Accuri™ C6 software (Version 1.0.264.21, Accuri Cytometers Inc., Ann Arbor, Michigan, USA).

### 4.5. Annexin V–Fluorescein Isothiocyanate (FITC) Staining Assay

Cell apoptosis was detected using the Annexin V–FITC Apoptosis Detection Kit II (Becton Dickinson, San Jose, CA, USA). Cells were seeded in a 6-well plate and treated with panobinostat for 48 h. Cells were treated with a mixture of annexin V binding buffer and PI in an appropriate proportion according to the manufacturer’s instructions. Cells were harvested, re-suspended in annexin V mixtures, and incubated at room temperature for 30 min in the dark. The samples were analyzed using the FACStar flow cytometer (Becton Dickinson, San Jose, CA, USA), and data were analyzed by BD Accuri™ C6 Software.

### 4.6. Western Blot Analysis

Protein expression was measured by Western blotting. Panobinostat-treated cells were harvested and suspended in lysis buffer (including a protease inhibitor cocktail and phosphatase inhibitor cocktail; Pierce, Rockford, IL, USA). Cell extracts were incubated on ice for 30 min and centrifuged at 12,000× *g* for 30 min at 4 °C. After centrifugation, the supernatant was collected, and the protein concentration was determined using a BSA Protein Assay Kit (Pierce, TX, USA). Proteins in whole cell lysates were separated using sodium dodecyl sulfate–polyacrylamide gel electrophoresis and transferred to polyvinylidene difluoride membranes (GE Healthcare Life Sciences, Buckinghamshire, UK). After blocking, membranes were probed with specific primary antibodies and incubated with peroxidase-conjugated secondary antibodies. The protein bands were visualized using a Chemiluminescent HRP Substrate (Millipore Corporation, Billerica, MA, USA) and an Amersham Imager 600 (GE Healthcare Bio-Sciences AB, Uppsala, Sweden). Chemiluminescence signal images were processed using the ImageJ software program, normalized to their loading controls, and the relative protein expression levels were compared with controls. The following antibodies were used: GAPDH, PARP, cleaved PARP, caspase-3, cleaved caspase-3, E-cadherin, MMP-9, phoshpo-Wee1, Myt1, Cdc2, Akt, phospho-Akt, FOXM1, and an anti-rabbit antibody.

### 4.7. Wound Healing Assay

Cells were cultured in triplicate for each concentration of panobinostat in a 6-well plate until achieving 80–90% confluency. Thereafter, the monolayers were scratched with a 200 µL pipette tip. The cell debris and medium were washed with fresh medium and replaced with a low serum medium containing increasing panobinostat concentrations (0, 10, 50, and 100 nM). The scratch area was observed and imaged using the Axiovert 135 microscope (Carl Zeiss, Oberkochen, Germany) at different time points (0, 24, and 48 h). The scratch area was measured by AxioVision Rel 4.8 software (Carl Zeiss), and the migration rate was calculated.

### 4.8. Xenograft Experiment

The animal experiments were approved by the Institutional Animal Care and Use Committee (IACUC#CBNU2017-0001, 3 January 2017) of Jeonbuk National University (IACUC of Jeonbuk National University) under the NIH guidelines (USA). Female SPF/VAF immunodeficient mice (4 weeks old) were obtained from Orient Bio (Dea Jeon, South Korea). The mice were allowed to acclimate to local conditions for 1 week prior to performing the experiments. Thereafter, 100 μL of Matrigel containing 5 × 10^6^ human GC cells (SNU484) were subcutaneously inoculated into the right flank of the mice. Following tumor implantation, the animals were separated into two groups: the untreated control group (*n* = 5, 10 μL DMSO) and the panobinostat-treated group (*n* = 5, 10 mg/kg in 10 μL of DMSO). When the tumor diameter reached 50 mm^3^, panobinostat was administered via an intraperitoneal injection on a daily schedule (5 days on and 2 days off) for the entire duration of the experiment. The animal experiment was terminated when tumors reached 2 cm in size. The tumor size was measured every 3 days using a caliper and calculated as width^2^ × length/2. Following anesthetization, the mice were sacrificed. Some tumor samples were fixed in 10% formaldehyde, whereas others were stored at −80 °C.

### 4.9. Statistical Analysis

All experiments were performed more than three times. The results were expressed as the mean ± SE. Student’s *t*-test and two-way analysis of variance with Duncan’s multiple range test were performed for statistical comparisons between groups. A *p*-value of <0.05 was considered to indicate a statistically significant result.

## 5. Conclusions

In conclusion, the molecular mechanisms of panobinostat treatment in GC cells involved the inhibition of cell growth, proliferation, metastasis, and cell cycle progression, which were mediated via the inhibition of Akt/FOXM1 activation and induction of apoptosis. Therefore, panobinostat could be a potential anticancer agent for the treatment of GC.

## Figures and Tables

**Figure 1 ijms-22-05955-f001:**
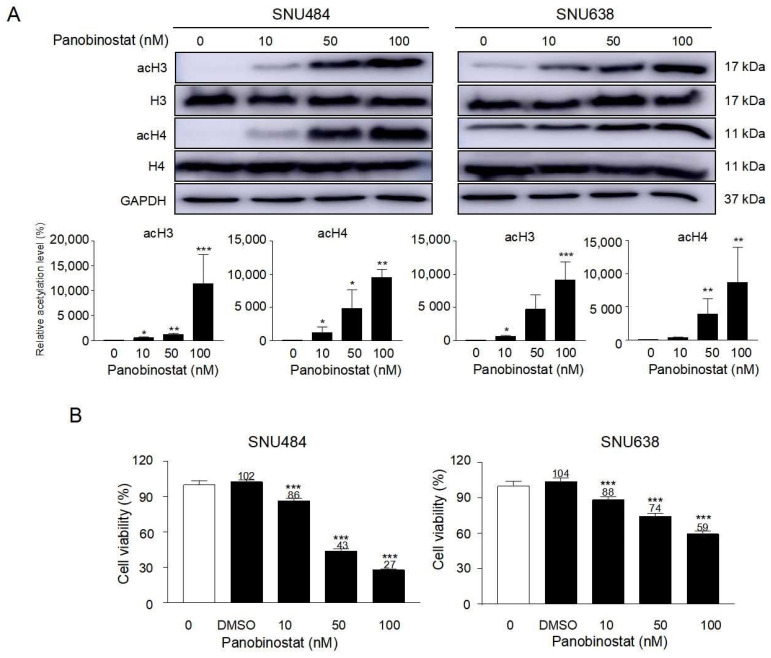
Panobinostat suppresses the proliferation of gastric cancer cells. (**A**) Human gastric cancer cell lines SNU484 and SNU638 were treated with panobinostat (0, 10, 50, and 100 nM) or DMSO for 48 h. (**A**) Histone acetylation was induced by panobinostat treatment in a dose-dependent manner, whereas no change in histone H3 and H4 in SNU484 and SNU638 GC cells was observed (0, 10, 50, and 100 nM). GAPDH was used as an internal control. The Western blot band intensity was quantitatively analyzed using ImageJ software (NIH, Bethesda, MD, USA). Data represent the mean ± SE of three independent experiments. * *p* < 0.05, ** *p* < 0.01, and *** *p* < 0.001 in comparison to the control. (**B**) Cell viability was measured by WST assays. Data are shown as the mean ± SD of more than three independent experiments with five dishes per experiment. *** *p* < 0.001 compared with the control. (**C**) A colony formation assay was performed using SNU484 and SNU638 cells treated with panobinostat (0, 10, 50, and 100 nM) to assess proliferation. Colonies were imaged. Data are shown as the mean ± SD of more than three independent experiments. *** *p* < 0.001 compared with the control. DMSO, dimethyl sulfoxide.

**Figure 2 ijms-22-05955-f002:**
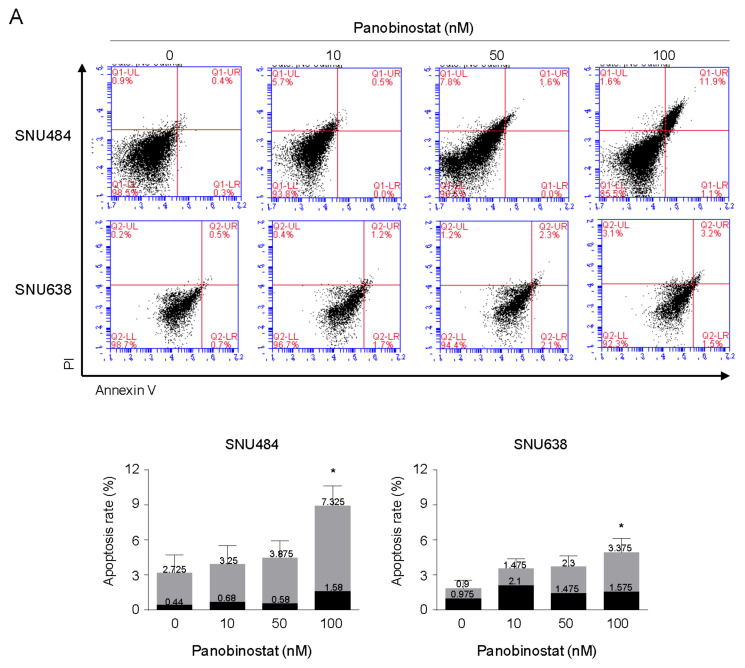
Panobinostat induces apoptosis in gastric cancer cells. SNU484 and SNU638 cells were treated with panobinostat (0, 10, 50, and 100 nM) for 48 h. (**A**) The induction of apoptosis was determined by annexin V–FITC/PI staining and flow cytometry analysis. The apoptotic regions were quantitatively analyzed. Data are shown as the mean ± SE of more than three independent experiments. * *p* < 0.05, compared with the control. PI, propidium iodide. Gray color; late apoptotic cells, black color; early apoptotic cells. (**B**) Western blot analysis of PARP, cleaved PARP, caspase-3, and cleaved caspase-3 in SNU484 and SNU638 cells harvested at 48 h after panobinostat treatment (0, 10, 50, and 100 nM). GAPDH was used as an internal control. The Western blot band intensity was quantitatively analyzed using ImageJ software (NIH, Bethesda, MD, USA). (**C**) Cell cycle distribution was detected by PI staining and flow cytometry. The proportion of cells in the sub-G1 phase was dose-dependently increased. Data represent the mean ± SE of three independent experiments. ** *p* < 0.01 and *** *p* < 0.001 compared with the control. FITC, fluorescein isothiocyanate; PARP, poly ADP-ribose polymerase; PI, propidium iodide.

**Figure 3 ijms-22-05955-f003:**
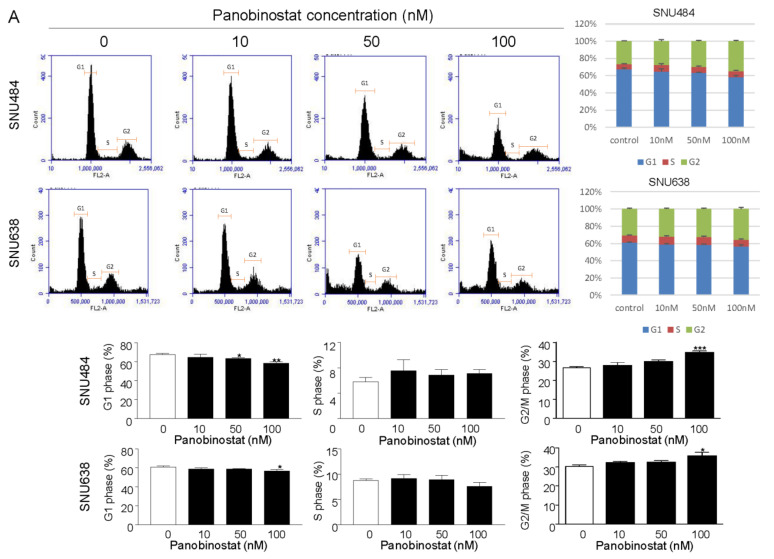
Panobinostat regulates cell cycle progression in gastric cancer cells. SNU484 and SNU638 cells were treated with panobinostat (0, 10, 50, and 100 nM). (**A**) Cell cycle distribution was detected by PI staining and flow cytometry and calculated as the percentage of cells in the G1, S, and G2/M phases. The scatter plots of the cell cycle show that the proportion of cells in the G1 phase was dose-dependently decreased by panobinostat. The fraction of cells in the G2/M phase was dose-dependently increased by panobinostat treatment. Data represent the mean ± SE of three independent experiments. * *p* < 0.05, ** *p* < 0.01, and *** *p* < 0.001 compared with the control. PI, propidium iodide. (**B**) Panobinostat affected the expression of cell cycle regulatory proteins. SNU484 and SNU638 cells were treated with panobinostat (0, 10, 50, and 100 nM) for 48 h. p-Wee1, Myt1, and Cdc2 levels were analyzed by Western blotting. GAPDH was used as an internal control. The Western blot band intensity was quantitatively analyzed using ImageJ software (NIH, Bethesda, MD, USA). Data represent the mean ± SE of three independent experiments. * *p* < 0.05, ** *p* < 0.01, and *** *p* < 0.001 compared with the control.

**Figure 4 ijms-22-05955-f004:**
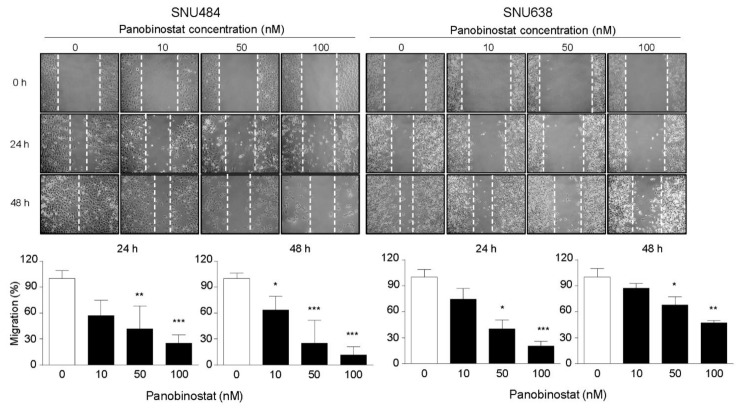
Panobinostat suppresses cell migration in gastric cancer cells. SNU484 and SNU638 cells were treated with panobinostat (0, 10, 50, and 100 nM), and migratory rates were detected by wound healing assays. Panobinostat significantly inhibited cell migration in a dose-dependent manner at 24 and 48 h. Data represent the mean ± SE of three independent experiments. * *p* < 0.05, ** *p* < 0.01, and *** *p* < 0.001 compared with the control.

**Figure 5 ijms-22-05955-f005:**
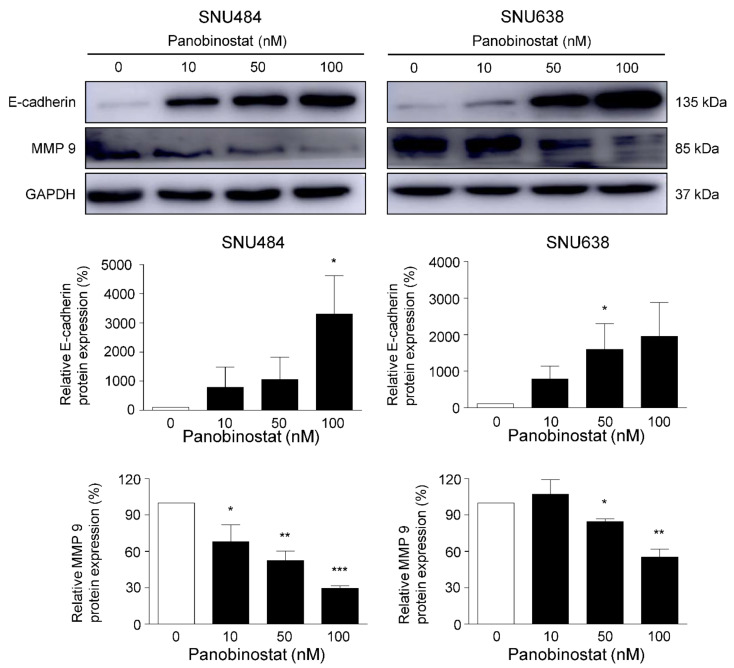
Panobinostat affects metastasis regulatory protein levels in gastric cancer cells. Protein levels of E-cadherin and MMP-9 in SNU484 and SNU638 cells were measured by Western blot after panobinostat treatment (0, 10, 50, and 100 nM) for 48 h. GAPDH was used as an internal control. The Western blot band intensity was quantitatively analyzed using ImageJ software (NIH, Bethesda, MD, USA). Data represent the mean ± SE of three independent experiments. * *p* < 0.05, ** *p* < 0.01, and *** *p* < 0.001 compared with the control.

**Figure 6 ijms-22-05955-f006:**
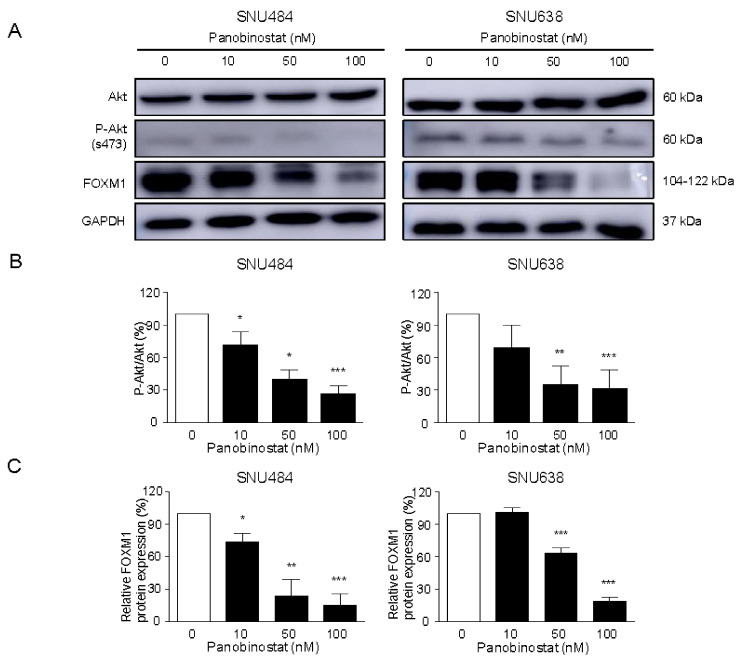
Panobinostat inactivates Akt/FOXM1 signaling in gastric cancer cells. SNU484 and SNU638 cells were treated with panobinostat (0, 10, 50, and 100 nM) for 48 h. (**A**) The expression levels of Akt, p-Akt, and FOXM1 were analyzed after treatment with panobinostat at different concentrations in SNU484 and SNU638 cells using Western blotting assays. GAPDH was used as an internal control. (**B**) Densitometric analysis result of p-Akt-to-Akt ratio from the Western blotting data shown in (**A**). The Western blot band intensity was quantitatively analyzed using ImageJ software (NIH, Bethesda, MD, USA). Data represent the mean ± SE of three independent experiments. * *p* < 0.05, ** *p* < 0.01, and *** *p* < 0.001 compared with the control. (**C**) The FOXM1 expression band intensity was quantitatively analyzed using ImageJ software. (**D**) Protein expression levels of Akt, p-Akt, FOXM1, and MMP-9 were detected by Western blot analysis after panobinostat treatment (50 nM) with Akt inhibitor (LY2940002) or Akt agonist (SC79). LY294002 (10 μM) and SC79 (10 μM) were pre-treated for 2 h and treated with panobinostat (50 nM) for 48 h. The Western blot band intensity was quantitatively analyzed using ImageJ software. Data represent the mean ± SE of three independent experiments. * *p* < 0.05, ** *p* < 0.01, and *** *p* < 0.001 compared with the control.

**Figure 7 ijms-22-05955-f007:**
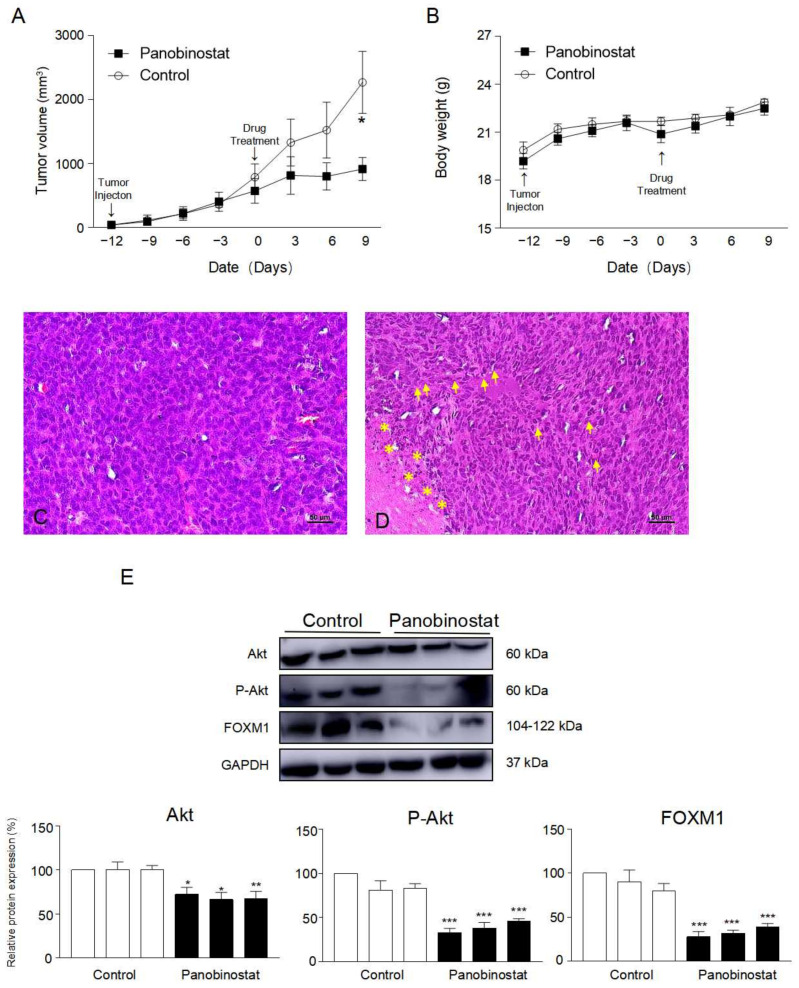
Panobinostat suppresses gastric cancer growth. Panobinostat inhibits the growth of gastric cancer tumors in xenograft mice; the results for vehicle-treated control and panobinostat-treated groups are shown (**A**). Tumor size was measured using a caliper every 3 days and calculated as width^2^ × length/2 throughout the experiments. Data represent the mean ± SE. * *p* < 0.05 compared with the control. (**B**) Animal body weights were measured every 3 days during the course of the experiments. (**C**,**D**) Histological examination of tumor tissue sections stained with hematoxylin-eosin was performed. (**C**) Control tumor tissue. (**D**) Panobinostat-treated tumor tissue. Bars = 50 µm; * indicates apoptotic and necrotic cells; the yellow arrow points to infiltrated lymphocytes. (**E**) The protein expression levels of Akt, p-Akt, and FOXM1 in animal tumor tissue were detected by Western blot analysis. (*n* = 3). GAPDH was used as an internal control. Relative protein expression levels (%) of Akt, p-Akt, and FOXM1 were calculated using ImageJ software (NIH, Bethesda, MD, USA). * *p* < 0.05, ** *p* < 0.01, and *** *p* < 0.001 compared with the control.

## Data Availability

The data presented in this study are available in article.

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
