# Peer review of "Inactivation of the Akt/FOXM1 Signaling Pathway by Panobinostat Suppresses the Proliferation and Metastasis of Gastric Cancer Cells"

_ijms, 2021, doi:10.3390/ijms22115955_

Round 1

Reviewer 1 Report

In the manuscript, the authors present a study regarding the effects of panobinostat on gastric cancer cells. The study is based on a culture cells experiment and an animal experiment. The authors used 10 mice that were divided in 2 group : the control group (5 mice) and the panobinostat treated group (5 mice). The results of the study are quite interesting. In my opinion it is a very good manuscript that can be published. Also, in order to improve the quality of the manuscript, please include at the end of the discussions part of the main manuscript some limitations of the study. I have also one question : Why do you used only female mice in the study? 

Author Response

Response: Your insightful comments are much appreciated. As you have suggested, we have added some of the limitations of our study to the end of the discussion.

The risk factors that drive sex disparity in esophageal cancer are unclear. Vizcaino et al reported that in esophageal adenocarcinoma the incidence rate in men is 6–10 times higher, and in esophageal squamous cell carcinoma the incidence rate is 2–3 times higher than in women (Vizcaino et al, 2002). Mouse models are important in the study of pathogenesis, therapeutic efficacy, and monitoring the progress of treatment in patients with esophageal squamous cell carcinoma (ESCC). Tumor xenograft models are widely used in cancer research due to the lack of immunity. The subcutaneous tumor xenograft model is often utilized as an animal model for ESCC research. (Lee et al, 2018). Some studies have reported using both xenograft and mice models for ESCC (Lee et al, 2018 and Yu et al, 2020). These studies concluded that subcutaneous tumor xenograft models are suitable due to their sensitivities toward chemotherapeutics for ESCC and are valuable research tools for studying tumor biology and disease mechanism for ESCC in an in vivo model. The effect of the experimental results due to gender differences has not been reported in ESCC xenograft studies. In this study, we used female mice as male mice tend to fight more often than females when placed together, and male mice have a more ferocious temperament when used for drug trials. Since this tendency would affect the experimental results, we used females. However, gender differences would need to be further studied in cancer and the associated mortality, as well as the therapeutic efficacy in ESCC xenograft animal studies.

Reviewer 2 Report

The authors analyzed the effect of the HDAC inhibitor Panobinostat in two gastric cancer cell lines and in xenograft mouse models.

Figure 1A. The increase in Histone acetylation upon Panobinostat treatment is intriguing. However, I recommend to use PanH3 or PanH4 as an additional loading control to control for total histone levels, since these may vary.

Please add the number of experiments that was used for the bar plot. Please indicate how the error bars were calculated. The label of the y-axis "relative protein expression" is somehow misleading since a posttranslational modification is analyzed, "relative acetylation level" might be more appropriate.

Fig. 1B, C: Please indicate the exact number of experiments that were performed.

For all bar plots, please show the individual data points underlying the bar plots (see PMID: 31884418). Please state which statistical test was used for the calculation of the p-values, please state if multiple testing correction was applied. Please show SD instead of SE.

Please provide the complete Western blots as a supplementary figure.

Please indicate how the proportion of late and early apoptotic cells was calculated.

Figure 3A: The proportion of cells in G2/M phase, although significant, differs only by 5-8% judged from the bar plots. Please rephrase the text accordingly.

Figure 4: Panobinostat decreased cell viability in the investigated cell lines. Can the effect of Panobinostat on cell migration be secondary - due to the decreased cell viability? Please comment.

Please give the product numbers of the antibodies used.

Minor:

line 361: "3rd distilled water" Please correct.

line 424: I assume, it should be "width2 x length/2"

Fig. 2B: western for c-PARP is missing for SNU484

Fig. 3: Western for GAPDH is missing for SNU484

Fig 7 C,D: HE stainings seem to be a bit flurry (at least on my screen). Please improve.

Author Response

Response: Thank you very much for your valuable comments. As you have suggested we have detected Histone H3 and H4 protein levels using western blotting after panobinostat treatment. We also have included the number of experiments that were used for the bar plot in Figure 1 in the revised manuscript. As you have also suggested, we have replaced the label on the y-axis from “relative protein expression” to “relative acetylation level” in Figure 1A.

Fig. 1B, C: Please indicate the exact number of experiments that were performed.

For all bar plots, please show the individual data points underlying the bar plots (see PMID: 31884418). Please state which statistical test was used for the calculation of the p-values, please state if multiple testing correction was applied. Please show SD instead of SE.

Response: Thank you for your valuable comments. For Figures 1B and 1C, the experiments were performed more than three times. As you suggest, we have now demonstrated the individual data points underlying the bar plots in Figures 1B and 1C. We used Student’s t-test and two-way analysis of variance with Duncan’s multiple range test for statistical comparisons between groups, and we have used SD instead of SE in the bar graph.

Please provide the complete Western blots as a supplementary figure.

Response: Thank you for your valuable comments. As you suggest, we have now provided complete western blot data in the Supplementary figures.

Please indicate how the proportion of late and early apoptotic cells was calculated.

Response: Thank you for your valuable comments. LR values indicate the percentage of early-stage apoptotic cells, and UR values indicate the percentage of late-stage apoptotic cells. We obtained the LR and UR values from three independent experiments and then calculated the average value.

Figure 3A: The proportion of cells in G2/M phase, although significant, differs only by 5-8% judged from the bar plots. Please rephrase the text accordingly.

Response: Thank you very much for your valuable comment. As you have suggested, we have now edited the percentage of the bar plots in the revised manuscript.

Figure 4: Panobinostat decreased cell viability in the investigated cell lines. Can the effect of Panobinostat on cell migration be secondary - due to the decreased cell viability? Please comment.

Response: Thank you for your valuable comment. We observed that panobinostat inhibited cell viability in ESCC cells. Panobinostat can affect cell migration through inhibition of cell proliferation. We believe panobinostat did not secondarily inhibit cell migration due to the decreased cell viability. We treated with mitomycin-c to inhibit cell proliferation to rule out this issue. Therefore, panobinostat is believed to suppress cell migration without affecting cell proliferation.

Please give the product numbers of the antibodies used.

Response: Thank you for your valuable comment. As you suggest, we have now included the product numbers of the antibodies in the revised manuscript. The following infromation ahs been included: Cell signaling technology: GAPDH (#2118s), H3 (#9717), H4(#2592), AcH3(#8173), Ach4(#2591), poly ADP-ribose polymerase (#9542), cleaved-PARP (#9541), caspase3(#9662), cleaved-casepase3 (#9664), e-cadherin (#5296), MMP-9 (#13667), phospho-Wee1 (#4910), Myt (#4282), Akt (#9272), phospho-Akt (#9271), anti-rabbit IgG (#7074), Santa Cruz: FOXM1 (SC-502), CDC2 (SC-163).

Minor:

line 361: "3rd distilled water" Please correct.

Response: Thank you very much for your valuable comment we have now corrected it.

line 424: I assume, it should be "width2 x length/2"

Response: Thank you very much for your valuable comment we have now corrected it.

Fig. 2B: western for c-PARP is missing for SNU484

Response: Thank you for your valuable comment we have now corrected Figure 2b.

Fig. 3: Western for GAPDH is missing for SNU484

Response: Thank you for your valuable comments we have now included the western blot data for GAPDH.

Fig 7 C,D: HE stainings seem to be a bit flurry (at least on my screen). Please improve.

Response: Thank you for your valuable comment we have modified the HE staining images in Figures 7C and D.

References

Vizcanio AP, Moreno V, Lambert R, et al. Time trends incidence of both major histologic types of esophageal carcinomas in selected countries, 1973-1995, Int J Cancer. 2002; 99:860-868

Lee NP, Chan CM, Tung LN et al. Tumor xenograft animal models for esophageal squamous cell carcinoma, J Biomed Sci, 2018; 25:66

Yu VZ, Ip JCY, Ko JMY, Tao L et al. Orthotopic xenograft mouse model in esophageal squamous cell carcinoma, Methods Mol Biol, 2020; 2129:149-160
